# Evaluation of Forward Osmosis and Low-Pressure Reverse Osmosis with a Tubular Membrane for the Concentration of Municipal Wastewater and the Production of Biogas

**DOI:** 10.3390/membranes13030266

**Published:** 2023-02-23

**Authors:** Mónica Salamanca, Laura Palacio, Antonio Hernandez, Mar Peña, Pedro Prádanos

**Affiliations:** 1Institute of Sustainable Processes (ISP), University of Valladolid, Dr. Mergelina s/n, 47011 Valladolid, Spain; 2Department of Applied Physics, Faculty of Sciences, University of Valladolid, Paseo Belén 7, 47011 Valladolid, Spain; 3Department of Chemical Engineering and Environmental Technology, University of Valladolid, Paseo Prado de la Magdalena 3-5, 47011 Valladolid, Spain

**Keywords:** municipal wastewater, forward osmosis (FO), low-pressure reverse osmosis (LPRO), tubular membrane, biogas production

## Abstract

Currently, freshwater scarcity is one of the main issues that the world population has to face. To address this issue, new wastewater treatment technologies have been developed such as membrane processes. Among them, due to the energy disadvantages of pressure-driven membrane processes, Forward Osmosis (FO) and Low-Pressure Reverse Osmosis (LPRO) have been introduced as promising alternatives. In this study, the behavior of a 2.3 m^2^ tubular membrane TFO-D90 when working with municipal wastewater has been studied. Its performances have been evaluated and compared in two operating modes such as FO and LPRO. Parameters such as fouling, flow rates, water flux, draw solution concentration, organic matter concentration, as well as its recovery have been studied. In addition, the biogas production capacity has been evaluated with the concentrated municipal wastewater obtained from each process. The results of this study indicate that the membrane can work in both processes (FO and LPRO) but, from the energy and productivity point of view, FO is considered more appropriate mainly due to its lower fouling level. This research may offer a new point of view on low-energy and energy recovery wastewater treatment and the applicability of FO and LPRO for wastewater concentration.

## 1. Introduction

There is a growing concern around the world over water scarcity in many countries, whereas the demand for fresh water continues to grow day by day. It is, therefore, necessary to optimize water resources for which the development of new technologies for wastewater treatment is essential. At the same time, and in the context of a circular economy, the use of renewable energy sources and obtaining new energy sources should be prioritized. In addition, the use of by-products such as acquiring nutrients for crops is also desirable. Actually, the production of clean water and nutrients as well as bioenergy is being actively promoted [1]. In this sense, urban Wastewater Treatment Plants (WWTPs) under anaerobic conditions are of great importance because they can deal with a high-quality effluent and allow the recovery of high-value products from the wastewater stream, such as nutrients, and allow for obtaining biogas from the organic matter present in urban wastewater [2].

Nevertheless, the recovery of nutrients and the obtention of energy through the digestion of sludge occurs currently with very low efficiency. Moreover, water is often reclaimed with expensive tertiary treatment. In order to reduce the costs and to improve the efficiency of municipal wastewater recovery, new technologies are desirable such as membrane technology, which is one of the most promising tools for water treatment fields [3,4].

Forward Osmosis (FO) is a membrane technology of growing interest for resource recovery from municipal wastewater. The high contaminant rejection and lower fouling of the forward osmosis membrane compared to pressure-driven membrane processes such as reverse osmosis, nanofiltration, and ultrafiltration make it a promising alternative for pre-concentrating municipal wastewater [5,6,7,8,9,10,11]. On the other hand, reverse osmosis technologies are now a fundamental part of the process of procurement of drinking water. Desalination technologies are devised to alleviate this global challenge and to avail freshwater that fits the desired purpose [12]. Desalination is defined as any process that is designed to remove dissolved salts and minerals from saline sources (seawater or groundwater) to make potable water for municipal (drinking and household), industrial, and agricultural (livestock irrigation) uses [13].

However, the reverse osmosis (RO) process still has numerous demands that should be fulfilled to optimize it, such as the reduction in operation and maintenance costs or the extension of the membrane’s lifetime. In the search to improve and satisfy the demands of the RO process, Low-Pressure Reverse Osmosis (LPRO) has been proposed. In LPRO, it is possible to work at low pressures and, thus, to save energy with the subsequent reduction in costs [14,15,16].

The FO process consists in the movement of water molecules from a low-concentration solution (feed) to a high-concentration solution (draw) through a semi-permeable membrane under an osmotic pressure gradient without the need to apply any hydraulic pressure [17,18]. In the LPRO process, the passage of water molecules from the feed solution to the other solution is due to the application of low pressures (3 to 5 bar) [14].

Both processes allow the concentration of municipal wastewater and, therefore, the concentration of organic matter while the extraction solution is diluted. Moreover, the previously concentrated water would be taken to an anaerobic digestion (AD) process to obtain biogas from the concentrated organic matter present in municipal wastewater, while in FO, the draw solution can be recovered by membrane distillation (MD) or RO to obtain high-quality water [5,19].

Membrane technology and, particularly, the advantages of the FO process and its different applications have given rise to numerous studies during the last decade. However, only some research uses real municipal wastewater directly in the FO process [4,20,21,22,23,24,25,26,27,28,29,30,31]. No references were found where LPRO was used alone to concentrate municipal wastewater.

The objective of this work was to study the behavior of a tubular membrane with an effective area of 2.3 m^2^ in two processes: FO and LPRO. In addition, we established a pilot-scale membrane system to concentrate real municipal wastewater from the Valladolid municipal WWTP into both FO and LPRO. Changes in membrane flux, reverse saline flux, fouling, as well as the concentration capacity of organic matter in each of the processes were also studied. In addition, the biochemical potential for methane production from concentrated municipal wastewater through both FO and LPRO processes at 35 °C was investigated. Moreover, both processes were compared and the influence of salt on methane production was evaluated.

In Table 1, some operational parameters are summarized for both LPRO and FO experiments.

## 2. Materials and Methods

This section shows a detailed description of the materials, procedures, and equipment used.

### 2.1. Forward Osmosis Membrane

In this study, a recently manufactured FO tubular membrane was used in collaboration with Berghof Membrane Technology GmbH (Berghof Membrane Technology Leeuwarden, Friesland, The Netherlands), a company that specializes in tubular membranes for the filtration and separation of industrial process streams and wastewater. The tubular membrane used was the TFO-D90 that contains 2.3 m^2^ of active surface area.

This module is 1.250 m long and has a diameter of 9 cm. It is provided with a PVC case that holds a multichannel tubular membrane with 118 channels with an average diameter of 5.3 mm. The channels consist in a porous support made of epoxy resin whose lumen is coated with Aquaporin Inside^®^. The membrane was designed to work in FO mode. However, it can also work in LPRO mode and, therefore, both modes were used in this work. The specifications of the membrane can be found in Table 2.

### 2.2. Experimental System: FO Mode and LPRO Mode

The FO system consisted of the membrane module, a draw solution (DS) compartment, and a feed solution (FS) chamber (Figure 1).

Feed and draw solutions were continuously co-current-recirculated through each flow channel by two pumps: a feed pump, Lowara AISI-304 multistage centrifugal pump model 5HM04P07M5HVBE (Xylem, Rye Brook, NY, USA); a draw pump, Sicce multi 5800 pump (Sicce, Pozzoleone, Veneto, Italy). Stainless-steel tanks with a maximum volume of 30 L were used and graduated to determine the volume that decreases in FS and increases in DS, which means water flux. In addition, conductivity meters: FS, sensION™ + EC71 (Hach, Loveland, CO, USA) and DS, Crison GLP 32 (Hach, Loveland, CO, USA) were used in both solutions. Conductivity was previously calibrated with NaCl. Volume flow rates were read by means of two rotameters to adjust the appropriate volume flow rate in each solution. To determine the inlet and outlet pressure of the system, two manometers were used at each inlet and two manometers at each solution outlet. The system has four valves to control the volume flow rates and pressures.

#### 2.2.1. FO Mode

In FO mode, the active layer of the membrane was made to face the FS and the pressure difference between FS and DS was kept equal to, or less than, 0.2 bar. The draw solution used in the experiments was NaCl (with different concentrations to be mentioned below) and, in all the experiments, concentration was kept constant by adding a saturated NaCl solution of approximately 5 M by means of a peristaltic pump Watson Marlow 323 (Wilmington, MA, USA) to maintain a constant osmotic pressure during the experiments. In addition, the draw was kept in agitation using a digital stirrer IKA EUROSTAR 100 Digital (IKA, Staufen, Germany).

#### 2.2.2. LPRO Mode

In the LPRO mode, the active layer of the membrane was made to face the FS and the pressure difference between the FS and DS was approximately 2.0 bar, regulating and controlling the pressure difference with the valves in the system. In the draw solution, there was only distilled water as, in this case, water passes from the FS to DS due to the applied hydraulic pressure difference applied on the feed side and not to the osmotic pressure difference as occurs in the case of the FO mode.

### 2.3. Membrane Characterization and Operation Conditions

To characterize the initial behavior of the membrane, the tests recommended by the manufacturer were carried out as shown in Table 2: deionized water vs. 1.0 M NaCl; 40 L/min as feed volume flow rate and 5 L/min as draw volume flow rate; temperature 298.15 K; Transmembrane pressure (TMP): 0.2 bar; FO-mode, batch process and co-current flow configuration. In addition, these tests were also carried out after each experiment, to check the performance of the membrane.

In addition, to be able to calculate the reverse salt flux $J_{s}$ and the water flux $J_{w}$, volume and conductivity data were taken in both tanks for each liter that was reduced on the feed side throughout the experiment. To calculate the flows, the same formulas of previous studies were used [32,33].

Equation (1) was used to calculate $J_{w}$, where $V_{FS\;t_{i+1}}$ and $V_{FS\;t_{i}}$ are the volumes permeated from the feed in times $t_{i+1}$ and $t_{i}$, respectively, and A is the surface area of the active side of the membrane.
(1)$$J_{w}=\frac{V_{FS\;t_{i+1}}-V_{FS\;t_{i}}}{A\left(t_{i+1}-t_{i}\right)}$$

Equation (2) was used to calculate $J_{s}$, where $C_{FS\;t_{i+1}}$ is the salt concentration of the feed solution in time $t_{i+1}$; $C_{FS\;t_{i}}$ is the salt concentration of the feed in time $t_{i}$.
(2)$$J_{s}=\frac{C_{FS\;t_{i+1}}V_{FS\;t_{i+1}}-C_{FS\;t_{i}}V_{FS\;t_{i}}}{A\left(t_{i+1}-t_{i}\right)}$$

Furthermore, only in the FO mode and with distilled water in the FS, different concentrations of NaCl were used as the draw solution to observe the changes in the fluxes and to be able to decide the most appropriate concentration. The concentrations used were 0.5 M, 1.0 M, 1.5 M, and 2.0 M and the volume flow rates *Q* were *Q_FS_* = 40 L/min in the FS and *Q_DS_* = 5 L/min in the DS, which are flow rates suggested by the manufacturer for quality checking, as mentioned in Table 2. The flow parameters were also changed to find the best speed ratio between the feed solution and the draw solution. The flows in the feed side, *Q_FS_*, were, in L/min, 20.0, 30.0, 40.0, and 50.0, and the flows in the draw side, *Q_DS_*, also in L/min, were 2.50, 3.75, 5.00, and 6.25, respectively, to maintain the same proportion between the FS and DS. To test different proportions between FS and DS ones, *Q_FS_* was set at 40.0 L/min with *Q_DS_* being 10.0, 5.0, and 2.5 L/min. This was performed for salt concentrations of 0.5 M and 1.0 M, which were the concentrations considered most suitable because they are similar to that of seawater, so in coastal areas, seawater could be used as a draw solution in order to reduce the costs of the process [34,35].

All measurements were carried out at a temperature of 298 K and the run time of the experiments ranged from 30 min to 60 min.

### 2.4. Municipal Wastewater Concentration

In this study, municipal wastewater from the outlet of the primary settler of the WWTP) of the city of Valladolid (Spain) was used to be concentrated in the membrane processes. The general characteristics of the municipal wastewater collected as measured are, on average, approximately 300.5 mg O_2_/L in Chemical Oxygen Demand (COD), 63.5 mg TOC/L (Total Organic Carbon), 0.96 g/kg in Total Solids (TS), and 0.45 g/kg in Volatile Solids (VS). It should be noted that the organic matter characteristics of the wastewater vary significantly depending on the day of collection, as the sample was not homogeneous over time and depends on various factors such as rainfall. The samples were kept in drums and refrigerated until use.

The following steps were followed in each concentration process:

#### 2.4.1. Initial Membrane Permeability

Before performing a test with municipal wastewater, a permeability test (tap water in DS and FS and with a pressure difference of 2.0 bar) was carried out to determine the permeability of the membrane. The volume flows rates were *Q_FS_* = 20 L/min and *Q_DS_* = 2.5 L/min, identical to those used in the concentration stage. Time and volume measurements were taken to calculate permeability. This was performed in triplicate and then the mean was calculated. This permeability was used to determine the initial conditions of the membrane before each process. These values allowed an optimum DS concentration to have a driving force for the FO process similar to that in LPRO.

#### 2.4.2. Wastewater Stabilization in the System

In the stabilization stage, the municipal residual water was added to the refrigerated FS tank, and its recirculating was continued for 1 h to stabilize and fill the entire system. On the DS side, only tap water was used. Volume flows rates were *Q_FS_* = 20 L/min and *Q_DS_* = 2.5 L/min.

#### 2.4.3. Concentration of Wastewater

A.Concentration by FO

Regarding the conditions of the membrane, 6 L of NaCl was used initially as draw solution with a constant concentration of 0.5 M, throughout the experiment. The FS volume flows rate was *Q_FS_* = 20 L/min, and *Q_DS_* = 2.5 L/min was set for the DS, to reproduce the flows of the membrane characterization and to spare energy. We started with 28.0 L of municipal wastewater in the FS tank plus the 8.0 L already in the system (from the previous stabilization step). Next, a volume of wastewater was added bit by bit to keep the volume in the FS tank constant, as, if the volume was allowed to decrease, bubbles would form, which could result in the loss of organic matter in addition to a phase change. A total of 42 L of wastewater was added. In total, 78 L of municipal wastewater was used in this stage and the process was stopped when 16 L remained, the minimum volume before bubble formation. The duration of the experiment was 5 h 46 min. A sample was taken at the beginning and the end of the concentration process to carry out a mass balance in organic matter.

For all FO concentration process, data on the change in volume and conductivity measurements were taken every minute to later be used to calculate the reverse salt flux $J_{s}$ and the water flux $J_{w}$.

B.Concentration by LPRO

The starting volumes in LPRO were 9 L of tap water in DS and 27.5 L of municipal wastewater in FS. Volume flow rates were the same as in FO (*Q_FS_* = 20 L/min and *Q_DS_* = 2.5 L/min). The pressure difference was approximately 2.0 bar (FS inlet 2.1 bar, FS outlet 2.0 bar, DS inlet 0.05 bar, and DS outlet 0.1 bar). We started with 29.0 L of municipal wastewater in FS plus the 8.0 L already in the system and added 41 L more to maintain the volume in the tank and use the same amount of municipal wastewater volume as in FO to be able to compare both processes. In total, 78 L of municipal wastewater was used in this stage and the process was stopped when 16 L remained, as in the FO concentration. The duration of the experiment was 9 h 40 min to work with the same volume of wastewater as in FO. Samples were also collected at the beginning and the end of the experiment and volume data were taken to calculate the water flux and conductivity—although, in this case, there was no reverse salt flux—to determine the salt concentration factor in FS.

#### 2.4.4. Membrane Washes

After each concentration test, 3 washes were performed. In each wash, tap water was placed in both tanks with recirculation and was left for 30 min with the same volume flow rates as before. The volume data of each tank were taken at the beginning and at the end and a sample was taken from each tank at the end of each wash to determine the remains of organic matter and to be able to make a mass balance. Subsequently, more washes were carried out with tap water until a conductivity of around 300 µS/cm, the conductivity of tap water, was obtained.

#### 2.4.5. Membrane Final Permeability

At the end, a permeability test was performed, with the same conditions as the initial (tap water in DS and FS and with a pressure difference of 2.0 bar) to ensure that there was no significant fouling, and the membrane recovered the initial conditions after the concentration test.

### 2.5. Biochemical Methane Potential (BMP) Tests

The anaerobic inoculum used was pre-incubated for five days at 35 °C to minimize its content of residual biodegradable organic matter. BMP tests were used to quantify the maximum methane production potential of concentrated municipal wastewater using the two membrane modes: FO and LPRO. BMP tests were performed at 35 °C. The concentrations of volatile solids (VS), total solids (TS), TOC, and pH in the inoculum and FO and LPRO substrates are shown in Table 3. The values for each substrate differ significantly due to the variability in the wastewater intake. Therefore, the initial value of the wastewater at the beginning and end of each process must be taken into account for comparison.

Here, 160 mL bottles (60 mL of working volume) were used for the BMP tests. The quantities weighed were approximately 25 g of inoculum and 35 g of the substrate samples. The bottles were sealed with a rubber septum and aluminum crimp caps, and then they were gassed with helium gas. Subsequently, they were taken to an orbital shaker model G10 Gyratory Shaker (New Brunswick Scientific, Edison, NJ, USA) in a hot chamber at 35 °C. All tests were performed in triplicate and the results were averaged. The biogas produced in the blanks (only with inoculum and with milli-Q water instead of WW) was subtracted from the biogas obtained in the other tests carried out with concentrated WW.

The specific methane production (SMP) was expressed in milliliters of methane produced per gram of TOC added (as substrate) (mL CH_4_/gTOC_subs_) (under normal conditions, *p* = 1 atm and T = 0 °C). 

### 2.6. Analytical Methods

Chemical oxygen demand (COD), Total Organic Carbon (TOC), Total Solids (TS), and Volatile Solids (VS) of municipal wastewater from WWTP were measured according to standard methods [36]. A Shimadzu (Nakagyo-ku, Kyoto, Japan) analyzer (TOC-L) was used to determine the concentration of TOC in the samples where COD could not be made, due to the interference of salts. However, salts also affect in the TOC analyzer and, for this reason, in some samples, dilutions for TOC measurement were necessary to decrease salt effects and to ensure that the measurement was correct. pH was measured using a pH meter pH Basic-20 Crison, (Hach, Loveland, CO, USA).

Anions (Cl^−^, SO_4_^2−^, PO_4_^3−^, NO_2_^−^, NO_3_^−^) were determined by High-performance liquid chromatography HPLC Waters (Milford, MA, USA) with conductivity detector 432, flow rate 2 mL/min, injection volume 20 μL, and oven temperature 25 °C.

The biogas (CH_4_) production in the BMP test was measured using a gas chromatograph Varian CP-3800 GC (Varian, Palo Alto, CA, USA) coupled to a thermal conductivity detector and equipped with a CP-Molsieve 5 A column (15 m × 0.53 mm × 15 μm) and another CP-Pora BOND Q column (25 m × 0.53 mm × 15 μm). A pressure sensor IFM PI 1696 (IFM Electronic, Essen, Germany) was used to monitor the pressure of the bottles.

## 3. Results

### 3.1. Characterization of the Membrane in FO

The characterization of the tubular membrane according to the manufacturer′s quality check gave results in accordance with the specifications provided by the manufacturer, and it is shown in Table 2. Averages of the results obtained are 5.99 L/(m^2^h) water flux, 0.33 g/(m^2^h) reverse salt flux, and 0.055 g/L specific reverse salt flux, $J_{s}/J_{w}$. However, there may be small deviations that can be attributed to the variability in each experiment or membrane manufacturing batch and other differences in measurement procedures. This part is important to determine the performance of this new tubular membrane and to be able to choose the most convenient parameters such as flow rate or draw solution concentration for the rest of the experiments. Moreover, as already mentioned, we need to determine the water flux in a range of concentrations and flows, both in the feed and draw sides, to find the optimal conditions to obtain similar drawing forces for FO and LPRO.

#### 3.1.1. Different Concentration of NaCl as DS

The experiments are carried out using NaCl as DS with concentrations of 0.5 M, 1 M, 1.5 M, and 2 M and volume flow rates of *Q_FS_* = 40 L/min and *Q_DS_* = 5 L/min, the same flows used for the quality check.

In all the experiments, we start with approximately 27 L of distilled water in FS and 6–7 L of NaCl solution in DS. An operation time greater than 30 min and lower than 50 min is reached in all the experiments. The concentration of the NaCl solution is kept constant in each case by adding saturated NaCl solution. Experiments are performed in at least duplicate from lower to higher concentration. The membrane is always washed with tap water to remove any salt that might remain in the system at the end of each experiment. In Figure 2, it can be seen how the permeate flow is greater as the concentration of DS increases. This was expected because the osmotic pressure difference is greater [37].

The reverse flux of salt also increases with increasing concentration of DS as can be seen in Figure 2. This is because $J_{s}$ is proportional to the concentration gradient between the two faces of the membrane [38]. Therefore, although using a high-concentration DS would allow a greater flow of water, it is not suitable for concentrating municipal wastewater when an anaerobic treatment is required, as the concentration of salts influences and can even inhibit the anaerobic process [22]. Furthermore, it is economically more profitable to use a lower concentration of NaCl as DS, and in addition to being cheaper, it has less salt reflux and is easier to regenerate [39]. For this reason, concentrations of 0.5 M and 1.0 M are used to characterize the membrane at different volume flow rates.

#### 3.1.2. Different Feed and Draw Volume Flow Rates

Experiments are carried out with different feed and draw volume flow rates using the two lowest concentrations studied, which are 0.5 M and 1.0 M NaCl as discussed in Section 3.1.1, and distilled water in FS. In the experiments, the operation time is greater than 30 min and less than 50 min. The objective is to determine the change in both the water flux, $J_{w}$, and the reverse salt flux, $J_{s}$, provided by the tubular membrane as the volumetric flow rates change. That is, to analyze the effect of polarization of the concentration due to the variation in the tangential velocity to the surface of the membrane. This study is carried out in two ways to allow the analysis of the importance of the tangential velocity in the FS and in the DS. First, both the DS and the FS vary, simultaneously keeping their difference constant as mentioned above in Section 2.3.

Subsequently, the study was performed by varying the flux difference between FS and DS. Thus, the FS volume flow rate was maintained at 40 L/min while the DS volume flow rate was varied from 10 L/min, 5 L/min, and 2.5 L/min.

As a general trend, it is observed that when the tangential velocity increases on both sides of the membrane (FS and DS) or only on DS, $J_{w}$ increases and $J_{s}$ decreases. Figure 3 shows the case in which the volume flow rates of the DS are varied, keeping the volume flow rates constant in FS. The increase in $J_{w}$ is due to the increase in osmotic pressure, which, in the FO process, is reduced because of the dilution of the salt concentration on the side of the membrane in contact with the DS because of the passage of water ($J_{w}$ itself). The increase in the tangential velocity in the DS reduces concentration polarization by increasing the salt concentration (and, therefore, the osmotic pressure) and $J_{w}$ [33]. The reverse salt flux, $J_{s}$, is proportional to the difference in concentration between both sides of the membrane, so it would be expected to increase with speed (as we saw in Section 3.1.1). However, in this case, the increase in $J_{w}$ produces a drag of the salt in the opposite direction that exceeds that produced by the difference in concentration, causing a decrease in $J_{s}$ with the tangential speed.

Normally, systems with low values of specific reverse salt flux $J_{s}/J_{w}$ (low $J_{s}$ and high $J_{w}$) are sought in FO. In our case, we see that increasing the speed, we achieve this effect. Figure 4 shows the effect of varying both speeds, keeping the ratio between them constant. It is clearly seen that the effect is greater when the concentration in the DS is higher. This is to be expected, because the concentration polarization on both sides of the membrane should decrease with rate more sharply when concentrations are higher. Finally, to analyze the relative importance of the tangential velocity in the DS, compared to FS, this effect is studied, keeping *Q_FS_* constant (see Figure 5). It is observed that the behavior is similar to that observed in the previous case.

If we analyze the relative decrease in $J_{s}/J_{w}$ in both cases (Figure 4 and Figure 5), considering the same range of *Q_DS_*, it is observed that they are very similar. This indicates that the tangential velocity on the DS side is much more important in maintaining a high value of $J_{w}$ (by keeping the osmotic pressure high), whereas, on the FS side, *Q_FS_* reduces the salt concentration value, which has a very little effect on $J_{s}$ as the salt concentration on the FS side is initially low.

However, it must be considered that, in this case, on the FS side, there is only pure water, as in real processes (such as the case of the concentration of wastewater), it is important to maintain a high tangential velocity in the side of the FS. This would prevent the accumulation of substances on this side of the membrane, facilitating mass transfer and preventing fouling.

Considering the results of the characterization performed, it is judged more appropriate to use a concentration of 0.5 M NaCl, for management and cost reasons. In addition, using this concentration, the osmotic pressure difference seems to be better controlled. This is because maintaining a constant concentration of 0.5 M in DS is easier and has less variability in permeate flux and is, therefore, more stable than using a 1.0 M concentration. Volume flow rates *Q_FS_* = 20 L/min and *Q_DS_* = 2.5 L/min are chosen. They are the lowest studied and maintain the same relationship as used in the quality check. Actually, there are no great differences if we increase the volume flow rate ratio in the permeate flow. Furthermore, it is easier to control our system from an operational point of view for these low speeds, as, otherwise, bubbles could be produced within the system, especially when working with wastewater. Moreover, the subsequent increase in the experiment timespan allows us to analyze in more detail the differences between the FO and the LPRO.

### 3.2. Flow Analysis in the Concentration Wastewater Processes: FO vs. LPRO

Figure 6 shows the evolution of the total volume of permeate with municipal wastewater in FS over time for both processes (FO and LPRO). The FO process in 5 h 46 min reaches the same permeate volume as the LPRO in 9 h 40 min. The driving force for FO comes from the concentration gradient due a concentration in the draw side C_DS_ = 0.5 M NaCl, while it is the pressure gradient ∆*p* = 2 bar that drives LPRO. It can be concluded that these driving forces have similar actions from the comparison of the fluxes through the membranes in both processes (FO and LPRO). In effect, the LPRO permeability, with a pressure gradient of 2, is (1.12 ± 0.66) × 10^−6^ m s^−1^, while, according to Figure 2, the water flux in FO, with C_DS_ = 0.5 M NaCl, is (1.20 ± 0.70) × 10^−6^ m s^−1^. Evidently, the osmotic pressure produced by C_DS_ = 0.5 M NaCl is sensibly higher, but their effects are greatly modulated by polarization dilution [33]. This is equivalent to the normalization of the driving forces.

Another important difference between both processes is the shape of the curve. Figure 6 shows that in the FO process, the behavior is completely linear (correlation coefficient (r_FO_ = 0.9991)), while in LPRO, the slope decreases slightly over time (although with a nice linear correlation coefficient (r_LPRO_ = 0.9974)). This means that in the case of the LPRO, the fouling is greater throughout the process. To analyze in more detail the difference in fouling in both processes, $J_{v}$ is determined by numerical derivation. Seven points are used to smooth the effects of experimental deviations. Figure 7 shows these values, which, in both cases, are divided by the initial flow to normalize them and facilitate comparison. In this graph, it is clearly seen that, in the LPRO process, the flow drop is much more drastic. It can even be said that in the case of FO, the flow remains constant throughout the process. To quantify fouling, it is possible to resort to analyzing the results to various models. In our case, we resort to the Hermia models with the following equation [40,41].
(3)$$\frac{J_{v}}{J_{v_{0}}}=\frac{1}{{\left(1+k\cdot t\right)}^{n}}$$
where *k* is the kinetic constant, and the exponent n depends on the fouling mechanism. In this case, as the FS molecules and particles are much larger than the membrane pores, it is very likely that the fouling mechanism is cake-type. To confirm this, the goodness of fit to the different models is analyzed for the case of the LPRO.

In the case of FO, the low decrease in flow versus time, together with the variability in the experimental values, makes it impossible to discriminate between the different mechanisms based on the goodness-of-fit results. The values obtained for the kinetic constant *k*, in Equation (3), by assuming a cake model (n = 0.5), are 0.003 ± 0.002 h^−1^ for FO and 0.143 ± 0.003 h^−1^ for LPRO, which confirm that fouling, in the case of LPRO, is much more important and fast.

Previous studies have concluded that fouling in processes in which pressure is applied is greater than in FO processes [42,43]. However, there is some discrepancy as, in some cases, fouling for RO and FO is compared with different membranes or parameters, so the results are not comparable [44]. In addition, there are no studies that compare LPRO and FO in membrane fouling using the similar conditions, so the results of this study provide new and interesting information for future studies. The results show that the fouling in LPRO is higher and faster than in FO as observed in Figure 7. However, for longer operating times, it can be expected that the fouling of the membrane, using real wastewater, will influence the membrane fouling also for FO as in some previous studies [4].

### 3.3. Concentration Process and Recovery

#### 3.3.1. FO Process

A volume of 78 L of municipal wastewater is passed through the FO unit and the flux variation and membrane fouling of this process are evaluated in Section 3.2. The mean permeate flux is 3.80 L/(m^2^ h), reducing the volume by 30%, and the mean reverse flux of salt is 0.63 g/(m^2^ h). The pressure difference of FS and DS is less than 0.1 bar. The anions and organic matter present in the wastewater before and after concentration with FO are shown in Figure 8.

The concentration of PO_4_^3−^ and SO_4_^2−^ increases 3 times, that is, FO reaches concentration factors (recovery, C_WWFO_/C_WW0_) of 3. This squares with the reduction in the volume of the experiment. Furthermore, this means that by increasing the expected concentration factor, the membrane has a good rejection for this type of anion. However, the chloride concentration increases from 120.00 mg/L to 551.48 mg/L, as shown in Figure 8, which is more than 3 times (right axes in Figure 8), which is due to the reverse flow of NaCl salt used as DS.

In this case, the TOC concentration in WW0 is 11.8 ± 1.2 mg/L and that in WWFO is 33.3 ± 1.1 mg/L, which is a very good recovery in TOC. Even the amount of organic matter recovered at the end of the experiment, and adding that of the 3 washings that were carried out, is somewhat greater than 100%. This indicates that, in the stabilization part, some of the organic matter remains adsorbed on the membrane and ends up coming out when carrying out the washings. The washing waters of both the FS and the DS experiments are analyzed to study if there is a high percentage retained in the membrane, if part of the organic matter is oxidized, or if it passes to the other side of the membrane and is collected in the DS. No amounts of TOC are observed in DS above the experimental error, so the tubular membrane has a good rejection against organic matter.

The mass balance is studied by osmotic washing of the pipes with water. Specifically, three water washes of 30 min each are performed. This is enough to recover any debris that may remain within the membrane. Therefore, after the FO concentration experiment from which initial and final samples are taken from both sides of both FS and DS, the three successive washes of 30 min are carried out and samples of both FS and DS are collected to analyze the evolution of the organic matter content. The total organic carbon (TOC) recovery results in FS obtained by the TOC analyzer are shown in Figure 9.

It is observed that in the concentration experiment by FO, 65.75% of the TOC is recovered, and with the subsequent three washes, the rest is recovered, reaching 100%. However, it should be considered that there appears a small excess of TOC (83.47 mg), over 100%, due to the adsorption on the membrane during stabilization (performed before the concentration process).

Other studies report losses in organic matter that may be because part of the organic matter remains adsorbed on the active surface of the membrane [21,27,30,32]. This does not occur in this work, because a stabilization stage is carried out and, therefore, we start from a TOC value that already has considered the possible concentration adsorbed on the membrane. Therefore, it can be concluded that there is a small amount of TOC that remains adsorbed on the membrane but ends up coming out with osmotic washes in the FO process.

Regarding the permeability tests before and after the FO concentration process and the washes, it is seen how the membrane recovers the initial permeability conditions after the concentration process, with the initial value being (5.31 ± 0.40) × 10^–12^ m s^−1^Pa^−1^ and the final value being (5.69 ± 0.12) × 10^−12^ m s^−1^Pa^−1^, both of the same order.

#### 3.3.2. LPRO Process

A volume of 78 L of municipal wastewater is passed through the LPRO unit and the flux variation and membrane fouling of this process are evaluated in Section 3.2. The permeate flow is 2.40 L/(m^2^ h), reducing the volume by 30% like in FO process. In this case, there is no reverse flow of salt, as DS is tap water and the osmotic pressure difference is given by the applied pressure difference of 2 bar. The anions and organic matter present in the WW before and after concentration with LPRO are shown in Figure 10.

As can be seen in Figure 10, the concentration of all anions (Cl^−^, PO_4_^3−^, SO_4_^2−^) increases approximately 3 times (right axis), that is, it increases by the same proportion as the volume decreases. This indicates that the membrane rejects the anions studied, not allowing them to pass from FS to DS.

In this experiment, the TOC concentration in WW0 is 54.1 ± 2.2 mg/L and that in WWLPRO is 141.6 ± 4.7 mg/L. In this case, the recovery of organic matter in the form of TOC, considering the washings that are carried out later and the mass balance of the concentration process, does not reach 100% recovery in TOC. The mass balance for the LPRO experiment is the same as in FO, the concentration experiment, and three washes of 30 min each with FS and DS sample collections. The total organic carbon (TOC) recovery results in FS obtained are shown in Figure 11.

The recovery results show that in the concentration experiment by LPRO, 55.56% of the TOC is recovered and, with the three subsequent washes, an additional 24.67% is recovered. Therefore, there is a 19.77% TOC loss. In total, 80.22% is recovered. In other similar studies of FO, they found losses between 10 and 20% of organic matter in FO processes due to the adsorption of organic matter on the membrane, as mentioned above [21,27,30,32]. This loss of organic matter by the LPRO processes, which has not been found in the literature, could be a consequence of the fact that in LPRO, there is greater fouling of the membrane and part of the organic matter remains adsorbed on the membrane matrix, as discussed in Section 3.3. It is well known that in processes where pressure is applied, fouling is greater [45,46].

Then, in this experiment of LPRO, there are losses of organic matter even when the system was previously stabilized. This may occur because by applying pressure, part of the organic matter remains adsorbed and, by spending more time in the experimental device, it can be partly degraded by the microorganisms present in WW.

Regarding the permeability tests before and after the LPRO concentration process and the washes, it is seen that the membrane has a significantly lower permeability at the end of the process of (4.74 ± 0.12) × 10^−12^ m s^−1^Pa^−1^ than the initial permeability of (5.6 ± 0.3) × 10^−12^ m s^−1^Pa^−1^, although both are still of the same order of magnitude. The permeability in LPRO drops more than in the FO process, which is once again because the LPRO process has greater fouling than FO and part of the organic matter remains in the membrane matrix as we have previously commented.

Although the analysis of the economic efficiency of the processes is not the objective of this work, it seems clear that the establishment of the pressure gradient, which is the driving force for LPRO, requires an extra consumption of power, because all the rest of the energy consumptions are common to both processes. Moreover, LPRO requires always longer times than FO to obtain equivalent permeated volumes.

### 3.4. Biogas Production

The methane production potential of the two different concentrated municipal wastewater substrates produced by FO and LPRO (WWFO and WWLPRO) is evaluated by means of BMP tests at 35 °C, a suitable temperature for biomethane production for 4 days [30,47,48,49]. Subsequently to day 4, production decreases over time. The pH before and after the BMP experiment is not altered, with the value being 7.4 ± 0.1 for WWFO and 7.5 ± 0.1 for WWLPRO. The methane production from concentrated water results to be 1800 ± 800 N mL of CH_4_/g of TOC for FO and 1600 ± 700 N mL of CH_4_/g of TOC for LPRO.

The results show that the concentration processes of both methods, considering the margin of error, have similar potential for methane production. However, it is observed that for longer times, the methane production capacity of WWLPRO and WWFO, as the experiment has been proposed, decreases with respect to the inoculum. This is a consequence of the fact that, in the case of the blank experience, the inoculum is mixed with pure water, while, in the other cases, it is mixed with the municipal WWs that have sulfates and chlorides usually present in the urban wastewater. Thus, for the WWLPRO, the concentration of Cl^−^ is 314.38 mg/L, and 345.68 mg/L of SO_4_^2−^, while for the WWFO tests, the concentration of Cl^−^ is 551.48 mg/L, and 433.93 mg/L of SO_4_^2−^. The higher concentration of chlorides in the case of WWFO is probably due to the saline flow characteristic of FO processes. This shows that for this type of application, it is important that the membrane had a low value for $J_{s}$.

Similar studies concluded that the increase in salinity in wastewater samples concentrated by FO had a negative effect on biogas production [22]. In addition, the abundance of sulphate reducing bacteria lowered the production of biogas in the anaerobic process [50,51,52].

A linear correlation is found between the amount of organic load and the biogas yield, with the increasing concentration of organic matter in the substrate being more detrimental to methane production [53,54,55].

Therefore, the inhibition processes in methane production due to salts might be considered in the design of this type of application. To improve long-term methane production, additional salt removal pretreatment steps may be necessary. For example, some processes for chloride removal may include precipitation of chlorides, oxidation, adsorption, and membrane separation processes. Chloride precipitation can occur in the form of CuCl or AgCl. Oxidation can be accomplished by ozone or by electrochemical methods Adsorbents and ion exchangers can also be used [56]. Regarding membrane processes for the removal of salts (such as chlorides), electrodialysis and reverse osmosis are possible techniques [57,58,59]. However, improvements of these processes, and subsequently more research, are required for future investigations.

## 4. Conclusions

Concentrating municipal wastewater by a tubular membrane in FO and LPRO processes is investigated in this research. It shows how the studied tubular membrane is suitable for both FO and LPRO processes as it has high water fluxes and low specific reverse salt flux. From the study of the membrane characterization carried out, we can select the appropriate work parameters to optimize FO. We select: *Q_FS_* (20 L/min), *Q_DS_* (2.5 L/min), and NaCl concentration (0.5 M) as DS based on productivity and costs. In addition, from the energy and productivity point of view and without considering the energy cost of the regeneration of the DS if needed, the FO process should be considered more appropriate attending to its lower fouling level in urban wastewater treatment. The TOC recovery results show how membrane washes are necessary to be able to recover the organic matter that remains adsorbed on the membrane or in the system. Regarding the methane production capacity, it is similar for the wastewater from both concentration processes (WWFO and WWLPRO). Although FO can slightly increase chlorides, the starting urban wastewater usually has a significant concentration of salts. For this reason, the inhibition processes in the production of methane are a factor to consider in the design of this type of application. Therefore, additional salt removal pretreatment stages would be needed to improve long-term production of methane.

## Figures and Tables

**Figure 1 membranes-13-00266-f001:**
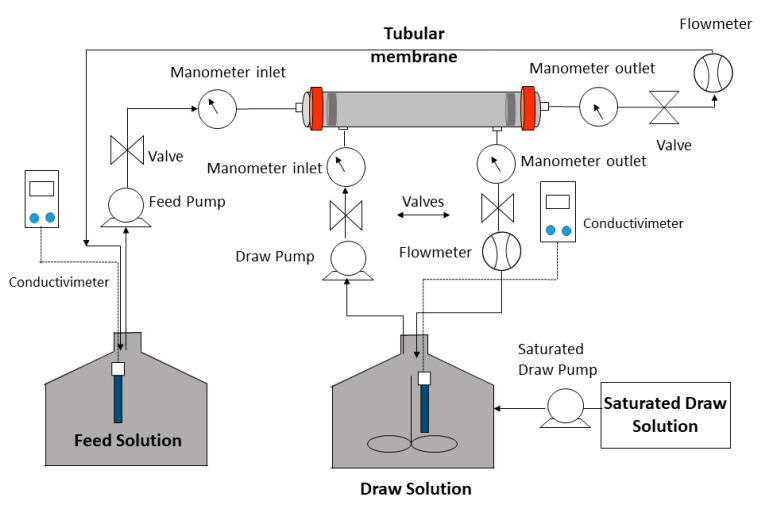
FO concentration system.

**Figure 2 membranes-13-00266-f002:**
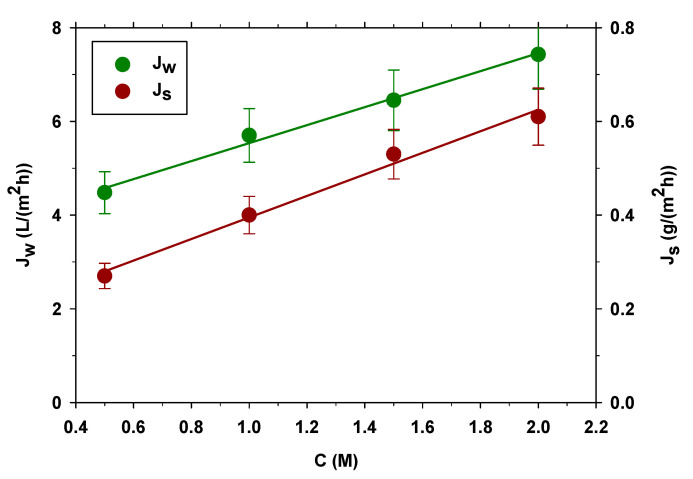
Average water flux (left side) and average reverse salt flux (right side) at different concentrations in the draw solution.

**Figure 3 membranes-13-00266-f003:**
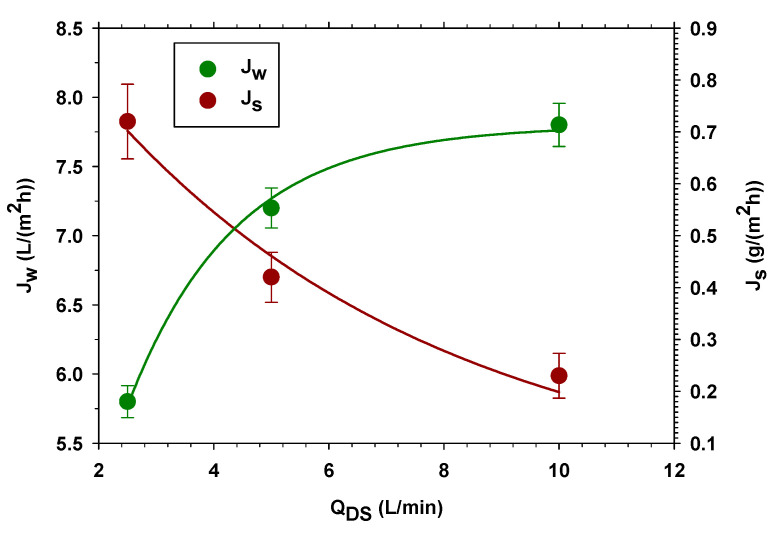
Water flux $J_{w}$ (left axis) and reverse salt flux $J_{s}$ (right axis) at different DS volume flow rates and 1.0 M NaCl draw solution.

**Figure 4 membranes-13-00266-f004:**
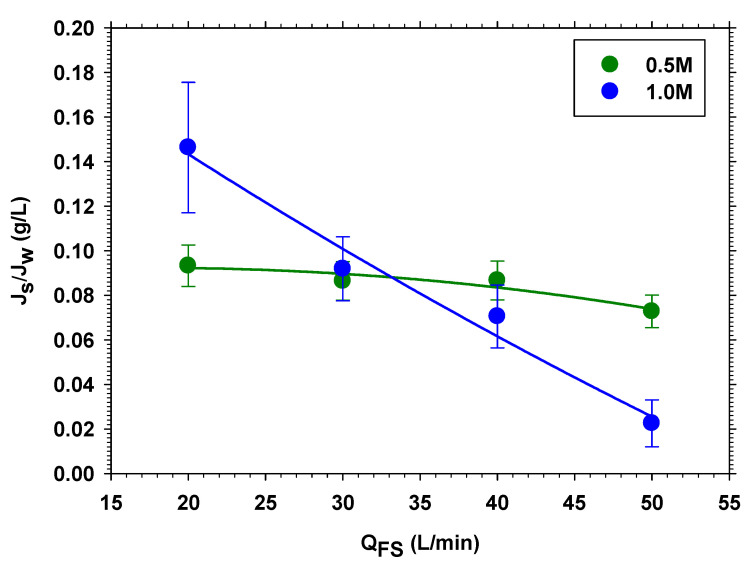
Specific reverse salt flux versus volume flow rate in FS, *Q_FS_*, for 0.5 and 1.0 M of NaCl.

**Figure 5 membranes-13-00266-f005:**
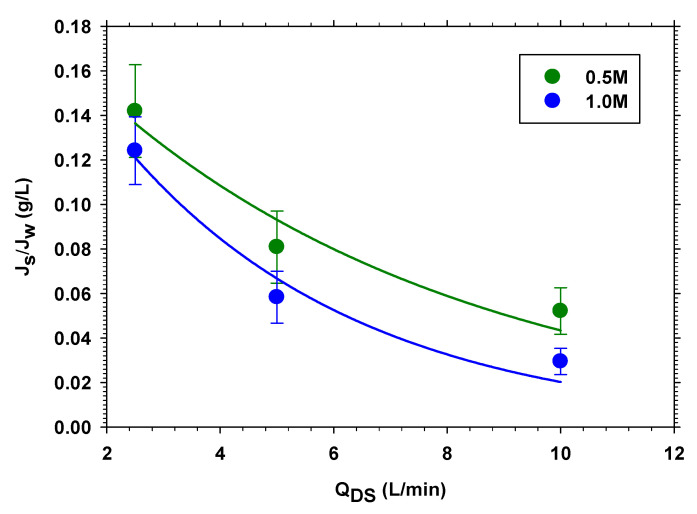
Specific reverse salt flux versus volume flow rate in DS, *Q_DS_*, for 0.5 and 1.0 M of NaCl.

**Figure 6 membranes-13-00266-f006:**
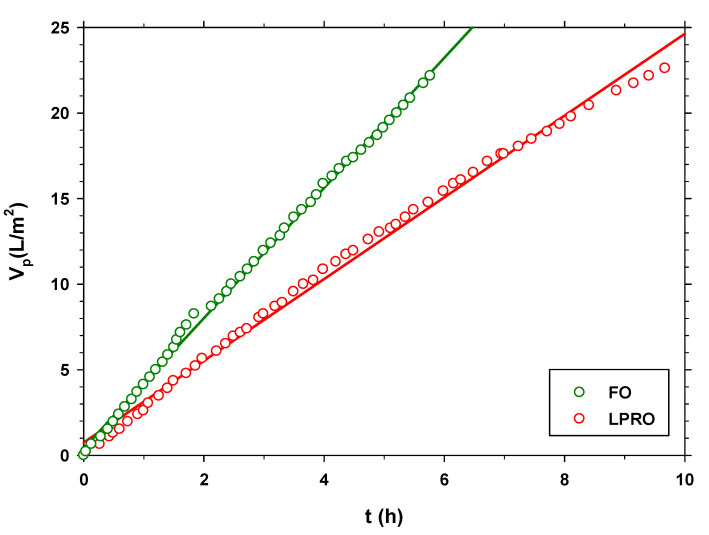
Vp versus t, for municipal wastewater in the FS with a recirculation volume flow rate of 20 L/min, comparing FO (C_DS_ = 0.5 M NaCl, *Q_DS_* = 2.5 L/min) and LPRO (∆*p* = 2 bar, *Q_DS_* = 2.5 L/min). A fit of the experimental points is added to a straight line to compare the linearity of both processes.

**Figure 7 membranes-13-00266-f007:**
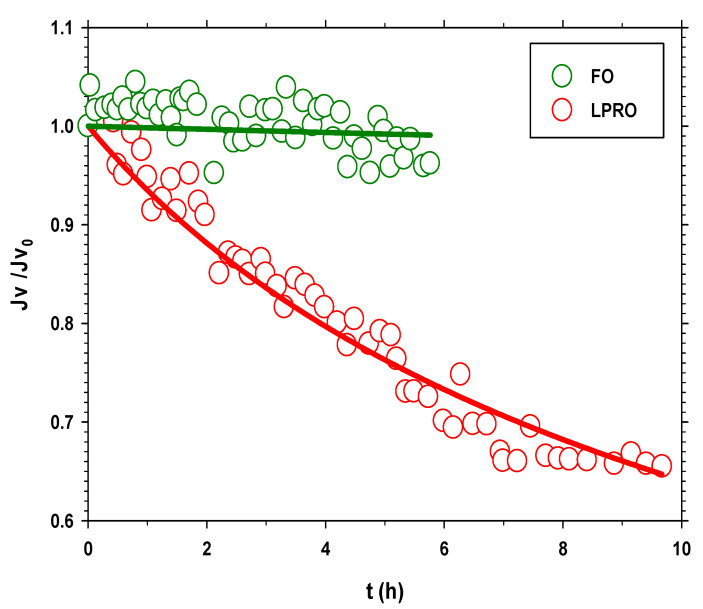
Permeate flow divided by the initial flow for the FO and LPRO processes of the previous figure (Figure 6). The solid line corresponds to the fit to the cake fouling model.

**Figure 8 membranes-13-00266-f008:**
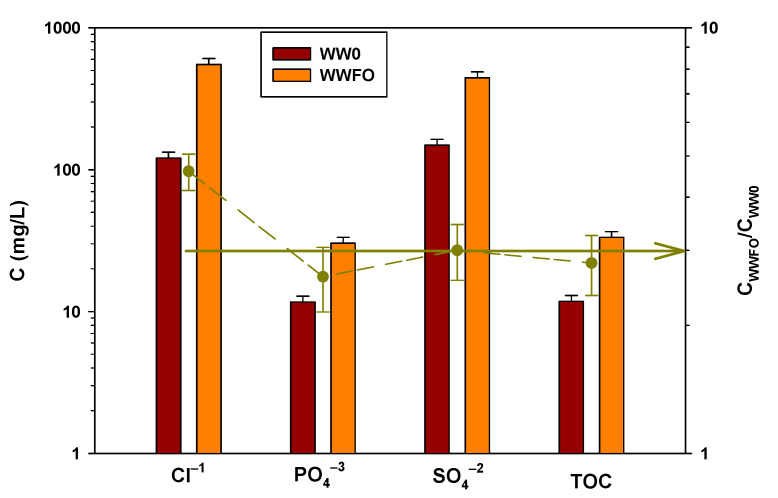
Concentration (left axis) and recovery (right axis) of Cl^−^, PO_4_^3−^, SO_4_^2−^, and TOC before and after the concentration process with FO.

**Figure 9 membranes-13-00266-f009:**
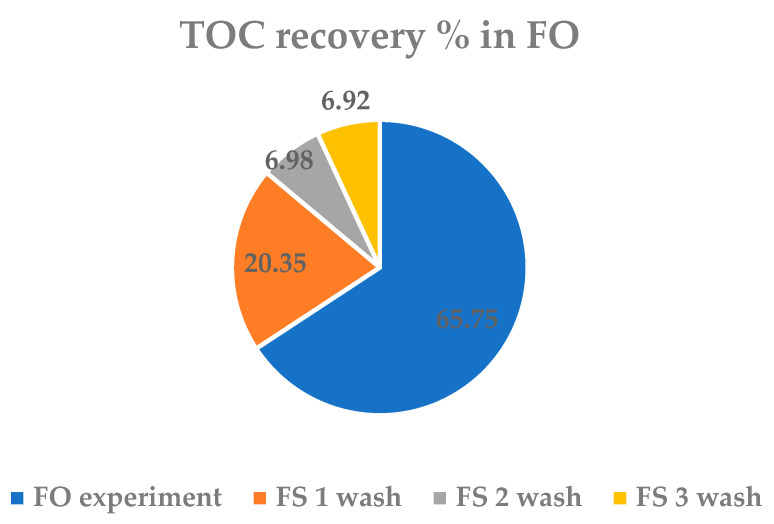
Total Organic Carbon (TOC) recovery in the concentration FO process.

**Figure 10 membranes-13-00266-f010:**
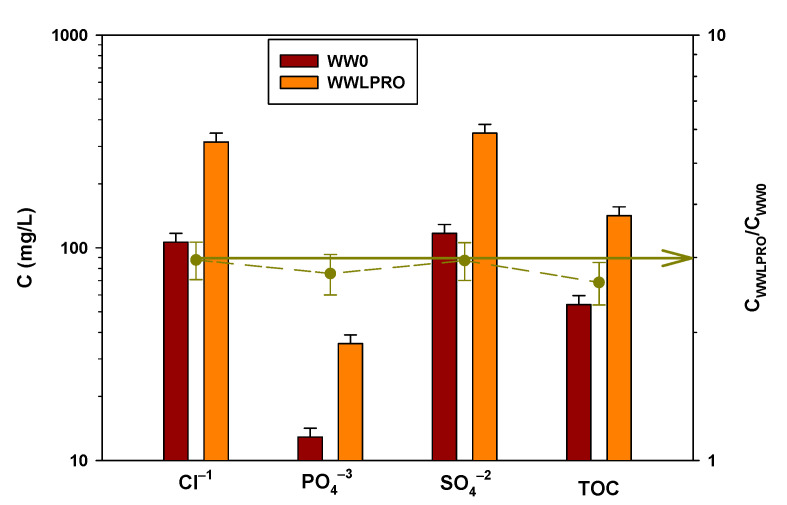
Concentration (left axis) and recovery (right axis) of Cl^−^, PO_4_^3−^, SO_4_^2−^, and TOC before and after the concentration process with LPRO.

**Figure 11 membranes-13-00266-f011:**
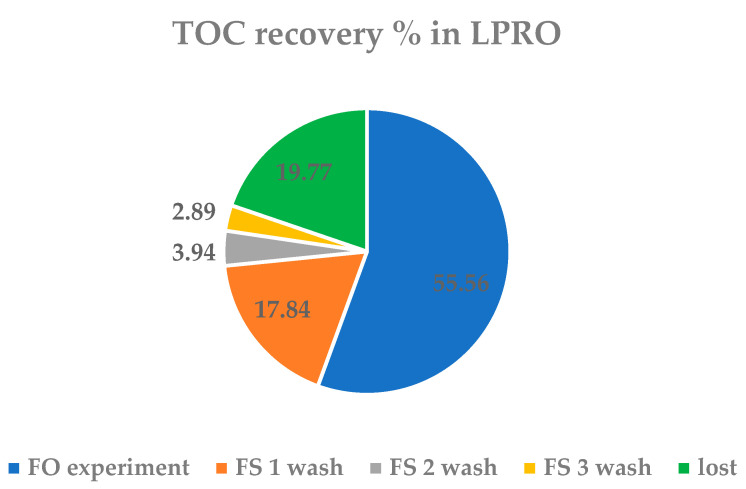
Total Organic Carbon (TOC) recovery in concentration LPRO process.

**Table 1 membranes-13-00266-t001:** Summary table with key operational parameters for both processes in concentrating municipal wastewater experiment.

		FO	LPRO
Operational parameters	Driving force	Similar in both processes	
	Volume of municipal wastewater treated (L)	78	
	Operation time (min)	346	580
	Concentration factor	3	
Parameters studied		Water flux (L/(m^2^h)) Reverse salt flux (g/(m^2^h)) Total recovery TOC concentration (included washes) (%) BMP test (NmL of CH_4_/g)	

**Table 2 membranes-13-00266-t002:** Specifications for the TFO-D90 tubular membrane module as provided by the membrane manufacturers.

Manufacturers	Berghof Membranes (Germany)
Membrane module	TFO-D90
Active area	2.3 m^2^
Average membrane tube diameter	5.3 mm
Lumen open cross-sectional area	26 cm^2^
Shell open cross-sectional area	15 cm^2^
Housing material	PVC-U
Potting material	Epoxy resin
Water flux (*)	>6 L/(m^2^h)
Reverse salt flux (*)	<1 g/(m^2^h)
Specific reverse salt flux (*)	<0.17 g/L

(*) Deionized water vs. 1.0 M NaCl, 40 L/min as feed volume flow rate and 5 L/min as draw volume flow rate, temperature 298 K, Transmembrane pressure (TMP): 0.2 bar, FO-mode, batch process, co-current flow configuration.

**Table 3 membranes-13-00266-t003:** Analyzed parameters of inoculum and substrates at the beginning of the BMP tests with standard errors.

	Volatile Solids (VS) (g/kg)	Total Solids (TS) (g/kg)	TOC (Total Organic Carbon) (mg TOC/L)	pH
Inoculum	10.60 ± 0.13	20.70 ± 0.21	828.11 ± 42.00	7.3 ± 0.1
FO substrate	5.70 ± 0.06	0.58 ± 0.01	33.30 ± 1.12	7.4 ± 0.1
LPRO substrate	21.88 ± 0.22	2.84 ± 0.10	141.60 ± 4.65	7.5 ± 0.1

## Data Availability

Not applicable.

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
