# Peer review of "Evaluation of Forward Osmosis and Low-Pressure Reverse Osmosis with a Tubular Membrane for the Concentration of Municipal Wastewater and the Production of Biogas"

_membranes, 2023, doi:10.3390/membranes13030266_

Round 1

Reviewer 1 Report

In general: The article poses an adequate experimental design and presents as a novelty to compare forward osmosis and low-pressure osmosis techniques in biogas production. However, it presents some formatting and expression errors that should be corrected, as well as some reasoning that should be clarified.

English:

Please revise grammar and use a spelling corrector to correct some errors (e.g. dimeter, absorbtion)

Some sentences should be split to facilitate understanding of the English language.

Use the correct tenses. Experiments, calculations, and design that have been performed must be expressed in past tenses. Reserve present tense for facts and procedures.

Units:

According to SI kilograms is expressed “kg”. Change to the correct form in all the article (figures and text)

Title:

Recommendation: Perhaps is better to change the sequence of techniques “Evaluation of Forward Osmosis and Low-Pressure Reverse Osmosis … 

Keywords:

Correct to: Low-pressure reverse osmosis

Abstract:

It must show the name of the membrane used.

The last sentence must be improved (…a new point of view on low-energy and energy recovery wastewater treatment…)

1.        Introduction:

The introduction is adequate, being, from my point of view, the part where the best care has been taken with the English expression.

Correct sentence in Line 79:

No references were found …

Here you have one reference, it may be others, please perform a thorough bibliographic search: Life cycle cost of a hybrid forward osmosis – low pressure reverse osmosis system for seawater desalination and wastewater recovery. Water Research1 January 2016.  R. Valladares LinaresZ. LiJ. S. Vrouwenvelder

2.       Materials and methods

Line 99: Suppress “model” not necessary and creates confusion.

Line 141: “tabulated” or “graduated”?

Line 148: Suppress “dissolution” or change to “solution” or “stream”

Line 190: “as an example given by the manufacturer…” !?

Line 204 “to be concentrated”

Table 2:

The bmp test à the BMP tests

(g VS /Kg) à (g/kg)  It is understood that it is Volatile solids. The same applies for others.

3.       RESULTS

Line 322, Table 1 shows specifications. You can see you meet them after showing them.

Figure 2 and 3: Jw units are not correct. Perhaps, L/(m2·h) or L·m-2·h-1? The same for Js

Figure 2: … at different concentration in draw solution.

Line 368-369: Check grammar.

Line 372: Why “this means”?

Line 383: “keeping the volume flow rates constant”. Where, in the FS?

Line 423: “using this concentration, the osmotic pressure difference seems to 423 control more efficiently the permeate flow of the system” Please, explain why or suppress this statement.

Line 440:  “in both these processes”?

Line 453: “the slope decreases over time (correlation coeffi-453 cient (rLPRO =0.9974))” This cannot be justified with a linear fitting (perhaps with an exponential fitting). If you want to justified this, the linear fitting t should be performed for the first points and then observe deviation.

Line 484: using the same conditions à similar conditions

Line 553: A volume of 78 L …

Line 582: Losses of organic matter seem to be very high if they are caused by adsorption on the membrane. Have they been quantified by an autopsy?

Perhaps, the explanation due to degradation by microorganisms is more plausible and should be emphasized.

Conclusions

Lines 635-641: Note that FO process would be more economic if seawater is used but if the draw solution must be regenerated the energetic advantage against LRPO is not clear. This should be discussed.

Line 645: “Therefore, additional pretreatment stages would be needed to improve long term production of me-645 thane.”

Which kind of pretreatments? Do they exist?

Reviewer 2 Report

The paper investigates concentration of municipal wastewater by tubular membrane in FO and LPRO process. Overall the scientific content is sound. To make it suitable for publication few suggestions follow:

1. The results are spread out and it is difficult for the reader to grasp it, it is suggested to develop a summary results table with key operational parameters for both processes and their results.

2. To have a better understanding of results the authors should present normalized data for the experiments, for example normalized permeate flux, normalized differential pressure and normalized ion rejection from the membranes for both processes. 

3. Figure 11 is for LPRO process not FO. Change figure caption.

4. The authors should include a discussion on the specific power consumption of both processes. 

Round 2

Reviewer 2 Report

The authors have made adequate changes, this article is now ready for publication.